# Quality of Life in Children with Prader–Willi Syndrome and the Impact of the Disease on the Functioning of Families

**DOI:** 10.3390/ijerph192316330

**Published:** 2022-12-06

**Authors:** Anna Rozensztrauch, Robert Śmigiel

**Affiliations:** Department of Nursing and Obstetrics, Division of Family and Pediatric Nursing, Wroclaw Medical University, 50-367 Wroclaw, Poland

**Keywords:** quality of life, Prader-Willi syndrome, family, child, rare disease

## Abstract

Objective: Prader–Willi (PWS; OMIM#176270) syndrome is a clinically distinct genetic disorder, caused by an abnormality in the 15q11-q13 region, referred to as the critical region. One of the most popular concepts existing in modern sciences, not only within psychology, but also in the aspect of all sciences that are related to human life and its course, is the quality of life (QoL). Though it is known that health-related quality of life in children with PWS can be reduced, less is understood about the impact on the family. We aimed to identify factors related to the quality of life of children with PWS and the impact of the disease on family functioning. Methods: A cross-sectional questionnaire survey. The subjects were 46 parents of children with PWS. The Computer Assisted Self-Interviewing (CASI) method was used; the Paediatric Quality of Life Inventory and the PedsQL Family Impact Module. Results: The PedsQL mean score was 49.0; (min–max: 5.6–90.8; SD = 16.8), with the highest scores in the Emotional Functioning (EF) (EF; 55.9; min–max: 5.0–100.0; SD = 22.0), and the lowest in the Social Functioning (SF) (SF; 42.7; min–max: 5.0–85.0; SD = 18.7) 56.4 (SD ± 14.7). The child’s age does not affect the quality of life, there were no statistically significant (*p* > 0.05). families have difficulties in performing daily activities (total score 27.6; SD 16.7), support family functioning (total score 28.9; SD 18.8) and effects physical domain (total score 27.7; SD 15.7). Conclusion: Research on the QoL of patients with PWS and their families is very important in order to assess the QoL, but also to provide the perspective of an active change in the perspective of a better treatment process, rehabilitation and communication in society.

## 1. Introduction

One of the most popular concepts existing in modern sciences, not only within psychology, but also in the aspect of all sciences that are related to human life and its course is the quality of life. Quality of life is a holistic concept, its definition includes three main aspects of human life perception: somatic, mental, as well as social. In medicine, its definition has evolved. Initially, the focus was on patient survival and prolonging life, but as medicine has progressed and interest in treatment effectiveness, cost of care and efficiency has increased, the focus has shifted to ensuring that patients live with dignity [1,2,3,4]. It has been pointed out that it is necessary to take into account the current physical, emotional and psychological, as well as social, well-being of the patient, which is a key determinant of his or her health status and understanding of quality of life, rather than simply relying on indicators to assess disease [5]. In the case of chronic diseases, such as genetic rare diseases, there is a reduction in quality of life, along with disability, depending on the type of disease. This is associated with a variety of difficulties and disadvantages, not only for people with Prader–Willi syndrome (PWS), but also for their careers [6,7,8]. Recently, it has become increasingly important to measure the QoL of rare diseases in children. To measure QoL, generic and disease-specific instruments may be applied to measure HRQoL in children and adolescents with the same rare diseases [9]. The generic instruments can measure QoL domains across diseases and can be used in different populations [10], the Pediatric Quality of Life Inventory™ (PedsQL™, Lyon, France) is one of the widely used instruments to measure QoL in children with disabilities.

Prader–Willi (PWS; OMIM#176270) syndrome is comprised of clinically distinct genetic disorders, caused by an abnormality in the 15q11-q13 region referred to as the critical region. PWS was initially described by Prader and Willi in 1956 [11,12] with diagnostic criteria revised by consensus in 1993 and revised again in 2001 to inform the clinical indications for genetic testing [6,13]. The roles of the genes responsible for the above syndrome are not fully understood, but they are known to contribute to neuronal development and function, and their absence affects the functioning of many systems during life. Approximately 60–70% of all PWS cases result from a de novo deletion on paternally inherited chromosome 15 in this region, and another 25–30% of PWS cases are associated with maternal single-stranded disomy of chromosome 15. The remainder are associated with mis-silencing and disruption of paternally inherited gene function in the q11-q13 region following defects, translocations, or inversions in the imprinting center located on the paternally inherited chromosome 15 [11]. The condition affects approximately 1/20,000–30,000 births [14]. Severe hypotonia manifests shortly after birth that leads to sucking and swallowing problems and delayed psychomotor development, partially improving with age. Characteristic facial features (narrow forehead, almond-shaped eyes, thin upper lip and downturned mouth) are often observed, as well as very small hands and feet. This initial phase is followed by the most characteristic signs, including an inability to feel satiety often leading to severe obesity in affected children as young as two years old [15]. The situation can rapidly worsen without proper care, leading to child endangering obesity. Other endocrine disorders associated with this syndrome contribute to the characteristic clinical picture of low growth due to growth hormone (GH) deficiency and impaired pubertal development. Regularly, decreased bone mineral density is detected in biochemical studies without alteration of calcium, phosphate, vitamin D or parathyroid hormone metabolism. The degree of cognitive dysfunction varies considerably from child to child. It is associated with learning disabilities and impaired speech and language development, which are further exacerbated by psychiatric and behavioral problems (tantrums, stubbornness, manipulative behavior, and obsessive–compulsive traits) [16,17]. Though it is known that these problems can reduce QOL in children with PWS [6,7], still less is understood about the impact of PWS on the family. Most of the research on PWS focuses on diagnostic problems and treatment, but there are few publications describing the impact of PWS on the everyday functioning of families. To date, there is limited information regarding the social, emotional, and financial burden on families of children with PWS. PWS is known to significantly impact on caregiver burden, has a negative influence on the QoL of the affected individual and the entire family and is a known source of stress for families [18,19]. Children with PWS report poorer QoL scores [20]. Certain aspects of QoL specifically physical and psychological aspects have been demonstrated to be lower in children with PWS when compared to both healthy and obese controls [21,22].

The study was conducted to assess the quality of life of children with Prader–Willi syndrome and the impact of the disease on family functioning. The specific aim was to study the influence of selected sociometric variables (child’s age, caregiver’s age, place of residence, caregiver’s gender and education, marital status) on the level of quality of life of children with Prader–Willi syndrome and on the level of family functioning, as well as to assess the influence of associated defects on the quality of life of children with Prader–Willi syndrome and on family functioning, and to measure the influence of the disease on caregivers’ marital relations and to estimate the correlation between the studied variables.

## 2. Materials and Methods

### 2.1. Study Group Characteristics

In the following research paper, the Computer Assisted Self-Interviewing (CASI) method was used. The group was selected on the basis of purposive selection. The study comprised 46 responders who were legal caregivers of children diagnosed with Prader–Willi syndrome, who were under comprehensive care at the Rehabilitation and Educational Center for Exceptional Children in Wroclaw and the Foundation “Potrafie Pomoc”. Parents of the children were informed before the study that participation in the study is voluntary and anonymous. The inclusion criterion was the consent of the patient’s legal guardian, a patient with a diagnosed disease entity (Prader–Willi syndrome). The criterion for exclusion from the study was lack of consent from the legal guardian.

### 2.2. Ethical Consideration

The study was conducted according to the guidelines of the Declaration of Helsinki and approved by the Institutional Review Board (or Ethics Committee) of Wroclaw Medical University (protocol code KB 76/2020 and 6 February 2020.

### 2.3. Measures

The research instruments included our own survey questionnaire, the study-specific questionnaire (SSQ), and two standardized instrument, the Pediatric Quality of Life Inventory (PedsQL™) 4.0 Generic Core Scales and the Pediatric Family Impact Questionnaire PedsQl 2.0 (PedsQL-FIM).

The SSQ, featuring family, and medical questions, was completed by participants. The SSQ also included the sociodemographic data of participants (e.g., age, sex, residence) and disease-related data (e.g., growth hormone treatment commencement, the prevalence of comorbidities).

The PedsQL Version 4.0 is a modular instrument designed to measures health in four domains: functioning in the physical sphere—8 questions (concerned problems with: walking more than one meter; running; participating in active play or exercise; lifting something heavy; taking a bath or shower by oneself; doing housekeeping; pain complaints; lack of energy); emotional functioning—5 questions (problems with: feeling anxious or scared; sad or depressed; angry; sleep problems; worrying about what will happen to him/her); social functioning—5 questions (problems with: having good relationships with peers; peers not wanting to be friends with him; teasing from peers; not being able to do things that peers can do; keeping up with peers) and school functioning—5 questions (problems with: paying attention in class; forgetting things keeping up with school assignments; missing school due to feeling unwell; missing school due to doctor appointments or hospital stays). It comprises 23 items to measure the QoL over the previous month in pre-school children (5–7 years old), primary school children (8–12), and adolescents (13–18). Results are recorded on a five-item Likert scale, where 0 stands for “never a problem” and 4 stands for “almost always a problem”. Responses are reverse-scored and linearly transformed to a 0–100 scale (0 = 100, 1 = 75, 2 = 50, 3 = 25, 4 = 0), where 100 indicates the best QoL. The total score is the sum of the average scores from each subscale. Lower scores indicate a poorer QoL [16,17,18,19,20,21].

The PedsQL 2.0 Family Impact Questionnaire (PedsQL-FIM). The 36-item PedsQL™ Family Impact Module is a parent-report instrument designed to assess the impact of pediatric chronic health conditions on parents and the family. It includes 6 subscales measuring parents’ self-reported functioning: Physical Functioning (6 items), Emotional Functioning (5 items), Social Functioning (4 items), Cognitive Functioning (5 items), Communication (3 items) and Worry (5 items); as well as 2 subscales measuring parent-reported family functioning: Daily Activities (3 items) and Family Relationships (5 items). The scale has five Likert response options, ‘never’, ‘almost never’, ‘sometimes’, ‘often’ and ‘almost always’ (corresponding to scores of 100, 75, 50, 25 and 0). Regarding the interpretation of the scale, higher scores indicate better functioning (less negative impact). The PedsQL™ Family Impact Module Total Scale Score is calculated as the sum of the 36 item scores divided by the number of items answered [23,24,25,26,27,28].

Scale internal consistency reliability was determined by calculating Cronbach’s coefficient alpha. Scales with reliabilities of 0.70 or greater are recommended for comparing patient groups, while a reliability criterion of 0.90 is recommended for analyzing individual patient scale scores. Importantly, the PedsQL 4.0 Generic Core Scales and the Pediatric Family Impact Questionnaire PedsQl 2.0 has been widely used across many chronic conditions of children.

### 2.4. Satistical Analysis

Statistical analysis was performed using Statistica 12 software (TIBCO Inc., Palo Alto, CA, USA). Arithmetic means, medians, standard deviations, range of variation (extreme values) were calculated for measurable variables. For qualitative variables, frequencies of their occurrence (percentages) were calculated. All studied variables of quantitative type were checked by Shapiro–Wilk test to determine the type of distribution. Determination of differences between groups was conducted using Kruskal–Wallis non-parametric ANOVA. Spearman’s rank correlation analysis between selected variables was also performed. An α = 0.05 level was used for all comparisons.

## 3. Results

Forty-six parents participated in the study. Most were from urban areas (80.4%) and had higher education (50.0%). The largest group among the children were those aged 2 to 4 years and 8 to 12 years and constituted 31.8% of the total group. The majority among the study children were female (65.5%). Growth hormone treatment was in place for 54.5% children and commenced at median age of 2 to 5 months. All children were living with two parents. The mean gestational age was 38.8 weeks (min–max: 35.0–42.0 weeks; SD = 2.3 weeks). The prevalence of comorbidities in study children occurred: heart defects in 25%, genitourinary in 50%, nervous system in 12.5%. The mean age of the parents was 38.7 years (min–max: 21.0–47.0 years; SD = 7.8 years).

### 3.1. Analysis of QoL Measured Using the PedsQLTM 4.0 Generic Core Questionnaire

For the PedsQoL-total score, the mean score was 49.0;(min–max: 5.6–90.8; SD = 16.8), with the highest scores in the Emotional Functioning (EF) (EF; 55.9; min–max: 5.0–100.0; SD = 22.0), and the lowest in the Social Functioning (SF) (SF; 42.7; min–max: 5.0–85.0; SD = 18.7) (Table 1).

The child’s age does not affect the quality of life; there were no statistically significant (*p* > 0.05) differences in the results obtained. In all age groups, the SF was the most affected. Although there is no statistically significant differences, the results indicated that the quality of life in the social domain decreases with child’s age ($\bar{x}$ = 52.9; SD = 17.8, for 2–4 years; $\bar{x}$ = 42.0; SD = 19,2, for 5–7 years: $\bar{x}$ = 37.9; SD = 18.2, for 8–12 years: $\bar{x}$ = 31.7; SD = 18.9, for 13–18 years: *p* = 0.157). Results are presented in Table 2.

Residence, and marital status had no significant impact on the children’s functioning in any of the analysed areas (*p* > 0.05). PF (*p* = 0.413), EF (*p* = 0.263), SF (*p* = 0.674), RF (*p* = 0.253). In relation to of associated comorbidities, low PF scores (total 44.9, SD ± 4.9) were found compare to others areas: EF (total 45.5, SD ± 10.4) and SF (total 45.8, SD ± 5.9).

### 3.2. Family Functioning in PedsQL-FIM

The results suggest that “worry” about the child’s future, medical treatment and how others will react to their child, affects families the most (total score 24.2; SD 21.7). Additionally, families have difficulties in performing daily activities (total score 27.6; SD 16.7), supporting family functioning (total score 28.9; SD 18.8) and the physical domain is affected (total score 27.7; SD 15.7). The PedsQL-FIM the average transformed score of the level of family functioning was 29.6 (SD 13.8). Results are shown in Table 3.

There was no significant impact on the level of family functioning from the parent’s sex (*p* = 0.742), age (*p* = 0.476) or education (*p* = 181).

## 4. Discussion

The functioning of a family with a disabled child brings with it a complexity of problems, not only in the perception of the quality of life of the parents as well as the children themselves, but above all in the impact of the illness on families. The family lacks adequate institutional as well as financial support. As many parents as there are, there may be as many opinions as to how to accept and cope with the uniqueness of their child, but an overburdened family system has a negative impact, not only on family ties, but also on marital relations between caregivers. Unfortunately, there is still a noticeable stigmatization in society of people with chronic illnesses, which are often accompanied by intellectual disabilities [20,29,30]. Hence, many parents are left without support. The diagnosis of PWS challenges parents to provide the best possible care. Although parenting is an integral part of many people’s lives, parenting a child burdened with a genetic defect, who may require multi-stage specialized treatment, is an additional burden on caregivers [31]. Family, in this case, is the pillar that gives the necessary strength to overcome the difficulties of daily life; if the attitude of caregivers is based on physical, emotional as well as social balance then it has a beneficial effect on the condition of the child client. In the study of Kayadjanian et al. [18], it was observed that caregivers experienced high caregiving burden. This result was highest in the group of families who are caregivers of adolescents and young adults. Caregivers reported that caring for a person with PWS negatively affected the relationship in their marriage, functioning outside of family life, and dramatically decreased sleep quality and mood. Based on previous research on the impact of Prader–Willi syndrome on family functioning, it is not possible to say conclusively on which factors caregivers’ attitudes and coping styles depend. Vitale’s study [32] revealed that the subjective key to coping with a stressful family situation, was the inevitable adaptation to the new situation to help the child with Prader–Willi syndrome to develop, while focusing on the needs of all other household members. The quality of life of children and family functioning with PWS has aspects that are not strictly related to the disease. The need for increased individual care for a child with a disability, depending on the child’s physical condition, social supports, and relationships within the family, significantly affects the well-being, psychological, socioeconomic, and social status of parents.

In a recent study in 2021, Meade et al. [33] found that PWS significantly affects the quality of life for both the affected child and the family. A particular group of caregivers who report feeling the greatest burden of raising a child with Prader–Willi syndrome are caregivers of children over the age of 12. Disruptions to routine, reduced social activities and social participation, as well as psychological difficulties have been reported to increase caregiver burden.

The analysis of our own research shows that children with chronic illness such as Prader–Willi syndrome have a reduced quality of life in terms of social, school, emotional and physical functioning. In the study by Whittington et al. [34], it was explained that the reduced quality of life in school and emotional spheres may be due to an inability to read emotions, with particular emphasis on facial expression. This relationship may be a direct cause of impaired relationships among peers. Obtained results indicate that in the opinion of caregivers as well as children, the lowest level of quality of life comes from functioning in the social sphere, while the emotional sphere, in the opinion of two groups of respondents, is the best rated. It was also noticed that children rated their quality of life better than their caregivers. A similar relationship in their work showed Wilson et al. [22] that, taking into account the aspect of functioning in a particular sphere, children responded more positively than caregivers. The greatest opposite attitude concerned the quality of life in the social sphere. The problem of the impact of the disease on family functioning concerns not only the caregivers but also the siblings, who are also an integral pillar of the family. In a study by M. Mazaheri et al. [20], it was unequivocally found that siblings of children with Prader–Willi syndrome reported difficulties in family functioning, communication problems and increased number of conflicts. Sibling illness was found to cause elevated levels of perceived behavioral stress, with higher than average feelings of depression, isolation, and anger.

The repetition of such a result in many studies dealing with the above-mentioned topics among researchers in Poland and abroad, supports the necessity of adapting the rehabilitation process to the requirements of everyday life and teaching simple skills, which are yet so crucial for independent existence. The literature also emphasizes the fact that more attention should be paid to psychological, social and material support for families diagnosed with PWS, which can be combined with the results of our own research and that of Thomson et al. [35] from 2017, in which parents clearly stated that despite the use of family and social support, the level of burden did not decrease. Results emerged that government agencies, service providers, family members and associations supporting children burdened with PWS and their families should provide practical and emotional support to assist caregivers in raising children with Prader–Willi syndrome.

The situation of people with Prader–Willi syndrome and their families, in the light of the above considerations, poses many challenges. The results of the above discussion indicate that PWS negatively affects family functioning. Parents unequivocally report fear for their child’s future, considering not only the management of further treatment, but also considering social acceptance. There is a relatively small number of researchers investigating the quality of families in which a child has PWS, and examining the impact of the disease on family functioning [36].

An important aspect of better family functioning as a whole is the relationship between individual family members. A special bond is the partnership between caregivers. For both parents, the birth of a child with a disability is a new situation to which neither of the partners has previously been able to adapt. In anticipation of the birth of a child with a disability they have had to quickly locate, obtain and navigate through a complex system of care, as well as realize the necessity of continuing to care for their child, even after the child has entered adulthood. Facing the complex needs of a child with Prader–Willi syndrome triggers distress that significantly affects the partners’ emotional state.

Research on the quality of life of patients with Prader–Willi Syndrome and their families is very important in order to assess the quality of life, but also to provide the perspective of an active change in the perspective of a better treatment process, rehabilitation and communication in society. A variable determinant in the objective assessment of quality of life for people affected by PWS is the character of the patient in question, his or her personality or situational factors. Individual character traits, temperament and personality determine the meaning that the affected person and their caregivers give to the disease [37]. The most difficult, and at the same time the most desirable, reaction is the acceptance of the situation by the caregivers, while feeling an active desire to provide rational help and support to their child. Authors studying the quality of life of children with PWS syndrome have found that there is no single way that guarantees satisfaction with life so far. This response is related to many factors including the degree of impairment, the number of comorbidities, age, gender, the number of children in the family, the quality of the marital relationship before the birth of the child, resilience to stress or relationships in society. Some limitation of the study should be considered. The first limitation is that QoL was assessed only from the parents’ perspective and child/self—report was not applicable. Parental reports may not fully reflect the subjective experience of their children. In the future studies, a child-specific questionnaire should be used and may focus on recruiting a larger sample to better understand source support parents of children with PWS. Consistent, overarching needs of parents and children may be identified or may vary from condition to condition. The second limitation is related to the fact that the assessment of the quality of life was carried out only in two centers in Poland. We have plans to involve more centers in our future research, both in Poland and abroad. Early multidisciplinary care can improve the quality of life not only for patients with PWS, but also for their families. The special role of health professionals is to provide support to families and tailor-made treatment, as well as to raise awareness among the general public about Prader–Willi syndrome and other rare diseases [38,39,40].

## 5. Conclusions

Many studies conducted in the world have shown that the quality of life of people with Prader–Willi syndrome is significantly at a low level, which is due to deficits in the functioning of various spheres of life. A decrease in the level of functioning is also noticeable in parents who act as caregivers for people with the above syndrome, which may be due to the need for constant care of the child. With the passage of time and medical advances, a holistic approach to the family as an inseparable whole has become the gold standard for service delivery in health care. In view of this, there is an increasing emphasis on the growing need for psychological support for parents of children with disabilities. It has been noted that impaired physical as well as mental health significantly affects the treatment process of children with PWS. The situation of parents of a child with a disability deserves tremendous support from society and loved ones in the family. Prader–Willi syndrome is an example of a disease in which early diagnosis and appropriate treatment play a key role, because it radically changes patients’ prognosis.

## Figures and Tables

**Table 1 ijerph-19-16330-t001:** Results of the PedsQL Generic Core.

PedsQL Domains	$\bar{x}$	Me	Min	Max	Q1	Q3	SD
Physical functioning (PF)	45.0	43.8	12.5	78.1	34.4	56.3	16.9
Emotional functioning (EF)	55.9	57.5	5.0	100.0	40.0	70.0	22.0
Social functioning (SF)	42.7	45.0	5.0	85.0	25.0	55.0	18.7
School/preschool/nursery (role)	52.3	52.5	0.0	100.0	45.0	60.0	19.3
Total score	49.0	50.5	5.6	90.8	41.3	58.6	16.8

$\bar{x}$, mean; Me, median; Max, maximum value; Min, minimum value; N, number of respondents; Q1, first quartile; Q3, third quartile; SD, standard deviation; PF, physical functioning; EF, emotional functioning; SF, social functioning; RF, role functioning.

**Table 2 ijerph-19-16330-t002:** PedsQL Generic Core scores by child’s age with PWS.

	Child’s Age (In Years)								*p* Value
	2–4		5–7		8–12		13–18		
	$\bar{x}$	SD	$\bar{x}$	SD	$\bar{x}$	SD	$\bar{x}$	SD	
PF	55.4	13.9	46.3	18.5	37.9	17.4	35.4	11.8	0.074
EF	68.6	19.9	69.0	11.9	40.7	19.9	40.0	15.0	0.061
SF	52.9	17.8	42.0	19.2	37.9	18.2	31.7	18.9	0.157
RF	59.5	24.3	54.0	13.4	45.0	21.4	50.0	5.0	0.845
Total score	59.1	16.4	52.8	12.2	40.4	18.1	39.3	8.3	0.201

$\bar{x}$, mean; Me, median; Max, maximum value; Min, minimum value; N, number of respondents; Q1, first quartile; Q3, third quartile; SD, standard deviation; PF, physical functioning; EF, emotional functioning; SF, social functioning; RF, role functioning.

**Table 3 ijerph-19-16330-t003:** Results of PedsQL Family Impact Module.

PedsQL-FIM Domain	$\bar{x}$	Me	Min	Max	Q1	Q3	SD
Physical functioning	27.7	25.0	8.3	58.3	12.5	37.5	15.7
Emotional functioning	33.4	35.0	5.0	70.0	25.0	40.0	15.7
Social functioning	29.1	25.0	5.0	65.0	15.0	40.0	15.5
Cognitive functioning	28.9	22.5	7.5	70.8	16.7	42.5	18.8
Communication	29.9	25.0	8.3	66.7	16.7	41.7	14.7
Worry	24.2	16.7	0.0	66.7	8.3	41.7	21.7
Daily Activities	27.6	21.9	6.3	62.5	12.5	37.5	16.7
Family relations	33.6	25.0	15.0	75.0	20.0	45.0	18.2
Parent HRQoL	29.4	24.3	11.3	64.0	20.6	40.3	13.4
Family Functioning	28.9	22.5	7.5	70.8	16.7	42.5	18.8
Total Impact Score	29.6	22.2	13.2	63.9	19.4	41.0	13.8

$\bar{x}$, mean; Me, median;Max, maximum value; Min, minimum value; N, number of respondents; Q1, first quartile; Q3, third quartile; SD, standard deviation; PF, physical functioning; EF, emotional functioning; SF, social functioning; RF, role functioning.

## Data Availability

Not applicable.

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
