# Peer review of "Quality of Life in Children with Prader–Willi Syndrome and the Impact of the Disease on the Functioning of Families"

_ijerph, 2022, doi:10.3390/ijerph192316330_

Round 1
Reviewer 1 Report
Thanks for the oportunity to review this interesting paper. This study aimed to assess the quality of life of children with Prader-Willi 68 syndrome and the impact of the disease on family functioning, and found that the quality of life score is highest in the Emotional Functioning, and lowest in the Social Functioning. The manuscript has some flaws to consider:.
1. In the abstract, the authors did not state clearly the objective of this study, and the conclusion sees to be divergent from the results.
2. In the introduction section, the first paragraph shoud be the introdcution of Prader-Willi syndrone, what is the problem, how big is the problem, then you can state the importanc of QoL in this disease.
3. In the introduction section, please give a review on the previous studies on this research topic, so that the audience can have an idea of what’s going on in this research area, please indicate the rationale for choosing these covariates.
4. In the Methods section ,please give a rationale for the sample size, why only 46 respondents
5. In the Methods section ,please give the psychometric properties of the measuremnt tools.
6. In the Methods section ,the authors used Spearman's rank correlation analysis, this needs to be checked whether it is approapriate.
7. In the discussion section, please do not repeat the results, instead just higlight the results and explain the results.
8. In the limitation section, please check whether recalling bias exists for this study.
Author Response
Dear Reviewer 1,
Thank you very much for sending us the consensus opinion about requested revision of our manuscript entitled: Quality of life in children with Prader - Willie syndrome and the impact of the disease on the functioning of families We appreciate the thoughtful comments and we have modified the manuscript in response to your suggestions, which we believe will further improve its quality.

Reviewer 2 Report
The authors describe a survey study investigating the quality of life of a child with PWS and the impact on family functioning. While there are various studies suggesting the low level of QoL of people with PWS, relatively there are fewer studies examining the impact of the disease on the level of family functioning, for which the results are expected to contribute further assessment of QoL and towards a better treatment or rehabilitation process. While the values of the current work is appreciated, there are some issues suggested to enhance the manuscript:
1.
The presentation and accuracy of the descriptive statistics need to be improved. For example,
ln 162, the item regarding “worry” in the questionnaire should receive a mean score of 24.2 with SD of 21.7 according to Table 3 on page 5. However, in ln 162, mean score of 24.2 with a SD value of 18.2 is reported, which has been mistaken with the values of family functioning.
Ln 166, the impact score should be 29.6 (SD = 13.8) instead of 29.06 (13.8), please check.
Ln 172, for the indication of age (p, 0476), is it referring to p = 0.476?
2.
Page 1-2, regarding the introduction, in addition to the background details of PWS, I think it would benefit to include the literature review on PedsQL modules have been employed in recent studies in relation to QoL, which would strengthen the framework and significance of the current study.
3.
Ln 71, regarding the stated variables to be examined in this study, is the result of caregiver’s gender presented and discussed in the relevant sections, or should it be mentioned here? More details of specifying the research questions would also benefit the section materials and methods. It seems the gender and some demographics of the caregivers are not given in the study group characteristics (ln 77), although the demographics of the children with PWS were given in the results section (ln 126-134).
4.
The discussion section should include a paragraph discussing the limitations of current study according to the cross-sectional nature and sampling of the current study, while the issues addressed here would be supplemented with the major findings of this work, so that further perspectives can be generated and extended to implementation of future studies.
5.
In the abstract, ln 8-13, the descriptions here are actually not describing the objectives. This part on “Objective” should be revised for explicit mentioning the objectives of the current study.
Author Response
Dear Reviewer 2,
Thank you very much for sending us the consensus opinion about requested revision of our manuscript entitled: Quality of life in children with Prader - Willie syndrome and the impact of the disease on the functioning of families We appreciate the thoughtful comments and we have modified the manuscript in response to your suggestions, which we believe will further improve its quality.

Reviewer 3 Report
Thank you for the opportunity to review the manuscript “Quality of Life In Children With Prader - Willie Syndrome And The Impact Of The Disease On The Functioning Of Families” for International Journal of Environmental Research and Public Health. Overall, it was thought provoking and enjoyable read. Generally speaking, I only have positive things to say about this research and don’t have any substantive concerns regarding the manuscript. I applaud the author(s) for undertaking such an important study. My admiration notwithstanding, I have concerns such that I am unable to recommend publication at this time. My concerns are outlined below, in no particular order with regard to the magnitude or consequence.
· Lines 78-86: Please embed information regarding IRB approval and oversight
· Lines 92-104: please include response options for questions. Did parents respond “yes” or “no” to occurrence of a problem? How were the questions combined? Please include reliability metrics for combined measures. Absent this information, I am unable to adequately review the results presented in the paper.
· Lines 105-115: Please include reliability metrics for combined measures.
· Discussion: Please discuss the limitations to the data and research presented in the paper.
Author Response
Dear Reviewer 3,
Thank you very much for sending us the consensus opinion about requested revision of our manuscript entitled: Quality of life in children with Prader - Willie syndrome and the impact of the disease on the functioning of families We appreciate the thoughtful comments and we have modified the manuscript in response to your suggestions, which we believe will further improve its quality.

Round 2
Reviewer 1 Report
No further comments. Thanks.
Author Response
Dear Reviewer,
Thank you very much.
Reviewer 2 Report
The authors have incorporated my comments in the previous review and the manuscript has been further improved.
I thank the team for adding the details in the introduction, especially the part on PWS to further illustrate the background of the study, which also address my previous comment 2. Please consider: (1) Ln 44-45: I suggest the sentences be rewritten as: To measure QoL, generic and disease-specific instruments may be applied to measure HRQoL in children and adolescents with the same rare diseases [9]. The information conveyed is obscured in the current writing. (2) Ln 84: QOL should be replaced by QoL as in consistency with other parts.
A further suggestion is on the added section regarding limitation: would the authors explain more for the statements in ln 306-312 regarding the second limitation, which is not clearly explained. Overall, the manuscript would be strengthened from further editing and checking on the proof.
Author Response
Dear Reviewer 2,
Thank you very much for sending us the consensus opinion about requested second round revision of our manuscript entitled: Quality of life in children with Prader - Willie syndrome and the impact of the disease on the functioning of families We appreciate the thoughtful comments and we have modified the manuscript in response to your suggestions, which we believe will further improve its quality.
REVIEWER COMMENTS 1
The authors have incorporated my comments in the previous review and the manuscript has been further improved.
I thank the team for adding the details in the introduction, especially the part on PWS to further illustrate the background of the study, which also address my previous comment 2.
Please consider: (1) Ln 44-45: I suggest the sentences be rewritten as: To measure QoL, generic and disease-specific instruments may be applied to measure HRQoL in children and adolescents with the same rare diseases [9]. The information conveyed is obscured in the current writing.
Thank you very much for you comment. We have incorporated suggested changes. Please see the following sentence:
To measure QoL, generic and disease-specific instruments may be applied to measure HRQoL in children and adolescents with the same rare diseases
REVIEWER COMMENTS 2
(2) Ln 84: QOL should be replaced by QoL as in consistency with other parts.
Thank you very much for your comment. We have incorporated suggested changes. Please see the following sentence:
Children with PWS report poorer QoL scores [20]. Certain aspects of QoL specifically physical and psychological aspects have been demonstrated to be lower in children with PWS when compared to both healthy and obese controls [21,22].
REVIEWER COMMENTS 3
A further suggestion is on the added section regarding limitation: would the authors explain more for the statements in ln 306-312 regarding the second limitation, which is not clearly explained.
The second limitation is related to the fact that the assessment of the quality of life was carried out only in two centers in Poland. We have plans to involve more centers in our future research, both in Poland and abroad.

Reviewer 3 Report
Thank you for the opportunity to review the revised manuscript “Quality of Life In Children With Prader - Willie Syndrome And The Impact Of The Disease On The Functioning Of Families” for International Journal of Environmental Research and Public Health. The authors, in an honest and professional way, addressed most of my concerns. Thank you. That said, one of my previously mentioned concerns remains: Please include reliability metrics for combined measures. Absent this information, I am (still) unable to adequately review the results presented in the paper.
Author Response
Dear Reviewer 3,
Thank you very much for sending us the consensus opinion about requested second revision of our manuscript entitled: Quality of life in children with Prader - Willie syndrome and the impact of the disease on the functioning of families We appreciate the thoughtful comments and we have modified the manuscript in response to your suggestions, which we believe will further improve its quality.
REVIEWER COMMENTS 1
Thank you for the opportunity to review the revised manuscript “Quality of Life In Children With Prader - Willie Syndrome And The Impact Of The Disease On The Functioning Of Families” for International Journal of Environmental Research and Public Health. The authors, in an honest and professional way, addressed most of my concerns. Thank you. That said, one of my previously mentioned concerns remains: Please include reliability metrics for combined measures. Absent this information, I am (still) unable to adequately review the results presented in the paper.
Thank you for your comments. Please see incorporated changes in the manuscript at the end of methods section
Scale internal consistency reliability was determined by calculating Cronbach’s coefficient alpha. Scales with reliabilities of .70 or greater are recommended for comparing patient groups, while a reliability criterion of .90 is recommended for analyzing individual patient scale scores.
Importantly, the PedsQL 4.0 Generic Core Scales and the Pediatric Family Impact Questionnaire PedsQl 2.0 has been widely used across many chronic condition of children.
